# Spirooxindole: A Versatile Biologically Active Heterocyclic Scaffold

**DOI:** 10.3390/molecules28020618

**Published:** 2023-01-07

**Authors:** Siva S. Panda, Adel S. Girgis, Marian N. Aziz, Mohamed S. Bekheit

**Affiliations:** 1Department of Chemistry and Physics, Augusta University, Augusta, GA 30912, USA; 2Department of Pesticide Chemistry, National Research Centre, Dokki, Giza 12622, Egypt

**Keywords:** spirooxindole, synthesis, natural product, alkaloids, biological properties

## Abstract

Spirooxindoles occupy an important place in heterocyclic chemistry. Many natural spirooxindole-containing compounds have been identified as bio-promising agents. Synthetic analogs have also been synthesized utilizing different pathways. The present article summarizes the recent development of both natural and synthetic spirooxindole-containing compounds prepared from isatin or its derivatives reported in the last five years. The spirooxindoles are categorized based on their mentioned biological properties.

## 1. Introduction

Spirocyclic compounds occupy a unique place within organic chemical compounds due to their rigidity and 3D-geometrical structure. A. Pictet and T. Spengler (1911) reported the first spiro-analog intermediate. Among all spirocyclic compounds, spiroindole-containing compounds represent an important branch of this class. This is attributed to the versatile biological properties established by diverse natural and synthetic analogs that may originate from the *C*-2 or *C*-3 indolyl ring with many heterocycles affording various motifs [1] (Figure 1).

Cipargamin (NITD609) and MI-888 are good representatives of these compounds which are under clinical studies for consideration as antimalarial and antitumor drugs, respectively. Cipargamin is capable of inhibiting the synthesized protein in *Plasmodium falciparum*. MI-888 is capable of inhibiting p53–MDM2 with high efficacy against human cancers [2]. MI-219 also revealed potent inhibitory properties against MDM2 protein with apoptosis induction in cancer cells and safe behavior towards normal cells. SOID-8 is an effective agent against melanoma capable of STAT3 (transcription 3) and JAK-2 (Janus-activated kinase-2) inhibition (Figure 2) [3].

The unique chemical and bio-properties of spiroindolyl-containing alkaloids have attracted the great attention of many researchers. Many review articles have mentioned the synthetic protocols and bio-properties associated with compounds possessing this scaffold within the last few decades [1,2,3,4,5,6,7]. This study summarizes the important recent development of either naturally isolated or synthetically prepared spirooxindole-containing compounds within the last five years (2018–2022) based on their biological properties.

## 2. Natural Spiroindoles Isolated from Plants and Microorganisms

The whole plant of *Flueggea virosa* is used in Chinese folk medicine against rheumatism, cephalic eczema, pruritus, and injuries. Flueindoline C **1** was isolated from the ripe fruits of *Flueggea virosa* (extracted by 95% ethanol and purified by silica gel column chromatography) [8]. Other spirooxindole derivatives **2** and **3** were isolated from the *Datura metel* L. seeds (extracted by 95% ethanol) (Figure 3). Spectroscopic techniques including UV, IR, 1D-, 2D-NMR, and mass spectroscopy (HRESIMS), in addition to electronic circular dichroism, supported the structures of **2** and **3**. Significant antiproliferative properties were observed compared to the reference drug 5-Fluorouracil (IC_50_ = 29.34, 24.3, 20.37; 18.97, 32.82, 47.63; 6.73, 19.38, 10.39 μM for compounds **2**, **3**, and 5-Fluorouracil against HepG2 “hepatoma”, MCF-7 “breast”, and SGC-7901 “adenocarcinoma” human cancer cell lines, respectively) [9].

Spirooxindole alkaloids **4**–**7** were isolated from the leaves of Malaysian *Mitragyna speciosa* (Kratom) (Figure 4). Promising binding affinities with *μ*-opioid were noticed by compounds **4** and **5** relative to that of Morphine (*K_i_* = 16.4, 109.8, 789.4, 1715.9, 4.19 nM for compounds **4**–**7** and Morphine, respectively) [10].

The stems and barks of *Nauclea officinalis* are traditionally used in China as folk medications due to their anti-inflammatory and antimicrobial properties. Spirooxindoles (2*S*,3*S*)-javaniside **8**, naucleoxoside A **9**, and naucleoxoside B **10** were isolated and identified as the active constituents of *Nauclea officinalis* (78% ethanol extract of the stem) (Figure 5). Mild inhibitory properties of nitric oxide production by LPS (lipopolysaccharide) in RAW 264.7 cells were revealed (% inhibition = 59.1, 76.6, 144.7, and 12.3 for compounds **8**, **9**, **10**, and Dexamethasone, respectively). However, no antimicrobial properties were observed against bacterial (*S. aureus*, *E. coli*, *K. pneumonia*, *P. aeruginosa*) and fungal strains (*C. neoformans* var. *gattii*, *C. albicans*, *C. tropicalis*, *A. fumigatus*). In addition, these compounds did not show antiproliferation properties (MTT assay, up to 100 μg/mL) against HepG-2, SKOV3, HeLa, SGC 7901, MCF-7, and KB carcinoma cell lines [11].

For a long time in China, the stem of *Gardneria multiflora* has been used for treating pain, herpes, eczema, snake bites, and food poisoning. Monoterpenoid indole alkaloids **11**–**14** were identified from the leaves and stems of this plant (extracted by 95% EtOH and purified by silica gel column chromatography) (Figure 6) [12].

Spirooxindole metabolites **15**–**19** were isolated from the marine fungus *Penicillium janthinellum* HK1–6 (Figure 7). None of the isolated compounds reveal considerable antibacterial properties against Gram-positive (*S. aureus*, *E. faecalis*, *E. faecium*) and Gram-negative (*E. coli*) bacterial strains [13].

## 3. Synthetic Spirooxindoles

### 3.1. Antibacterial and Antifungal Spirooxindoles

A variety of 3-spirocyclopropyl-2-oxindoles **21** were synthesized through the methyleneindolinones **20** with the appropriate aromatic aldehyde and TsNHNH_2_ in MeCN. The reaction was assumed to take place via the formation of the corresponding hydrazone due to the reaction of an aromatic aldehyde with TsNHNH*_2_* which afforded the aryl-diazomethane in presence of K_2_CO_3_. The latter due to the interaction with **20** finally furnished the spirooxindoles **21** (Figure 1). Some of the targeted agents showed promising antibacterial properties. The most potent was **21d** (R = Br, Ar = 3-C_6_H_4_-O-CH_2_-Ph) revealing considerable antimicrobial properties relative to Ciprofloxacin against Gram-positive (MIC = 0.49, 0.007; 0.24, 0.007 μM for **21d** and Ciprofloxacin against *S. pneumonia* and *B. subtilis*, respectively) and mild behavior against Gram-negative bacteria (MIC = 7.88, 6.88; 3.9, 0.49 μM for **21d** and Ciprofloxacin against *P. aeruginosa* and *E. coli*, respectively) [14].

A series of spiro[indoline-3,4′-pyrans] **27**/**28** was synthesized via multicomponent reaction of 5-sulfonylisatins **22**, malononitrile/ethyl cyanoacetate **23**, and *β*-ketoester/*β*-diketone **25/26** in stirring methanol containing piperidine (basic catalyst) at room temperature (Figure 2). The reaction was assumed to proceed through 3-cyanomethylidene-2-oxindoles **24** intermediacy followed by the addition of **25**/**26**, finally affording the corresponding spiro-compounds **27**/**28**. Independent synthesis of the spiro derivatives **27/28** through the reaction of **24** with **25/26** under similar reaction conditions supported the proposed reaction sequence. Some of the synthesized agents revealed promising antibacterial (Gram-positive “*E. faecalis*, *S. aureus*” Gram-negative “*E. coli*, *S. typhi*”) and antifungal (*C. albicans*, *A. brasiliensis*) properties comparable with the standard referenced drugs Tetracycline and Amphotericin. Compounds **27f,h** and **28a,c,e,f,g** are the most potent against *Staphylococcus aureus* with considerable gyrase inhibitory properties (IC_50_ = 18.07–27.03 μM) relative to that of Ciprofloxacin (IC_50_ = 26.43 μM) [15].

The spiro[indoline-3,2′-[1,3,4]oxadiazols] incorporated pyridinyl heterocycle **30** was obtained through cyclization of the appropriate hydrazones **29** in refluxing acetic anhydride (Figure 3). Considerable antibacterial (*B. Subtilis*, *S. aureus*, *E. coli*, *S. typhi*) properties were noticed by the synthesized agents relative to Gentamicin. Additionally, antifungal (*C. albicans*, *C. oxysporum*, *A. Flavus*, *A. niger*) properties were exhibited by the spiro-compounds compared with Fluconazole. The most effective agent synthesized was that with R = Cl (MIC = 12.5 μg/mL against *B. subtilis* and *E. coli* for both the effective agent synthesized and Gentamicin; MIC = 12.5 μg/mL against *C. oxysporum* for both the effective agent synthesized and Fluconazole) [16].

Spiro-β-lactam-oxindoles **31** were obtained through [2+2] cycloaddition of isatin-imines with aryloxy acetic acid (Figure 4). All the synthesized agents revealed weak antibacterial properties (>200 μM) against *E. coli*, *P. aeruginosa*, and *S. aureus* [17].

The spiro[indoline-3,3′-pyrazoline]-2-ones **33** and spiro[indoline-3,4′-pyrimidin]-2-ones **34** were obtained through reaction of 3-(2-oxo-2-ethylidene)-2-indolinones **32** with hydrazine hydrate in refluxing ethanol containing Et_2_NH or thiourea in refluxing ethanolic KOH, respectively (Figure 5). Promising antibacterial properties were noticed against *B. subtilis* (MIC = 0.348–1.809 mM) and *S. aureus* (MIC = 0.044–0.226 mM) relative to Imipenem (standard reference, MIC = 0.026, 0.026 mM against *B. subtilis* and *S. aureus*, respectively) [18].

Analogously, a series of **34** was obtained through a multicomponent reaction of phenacylidenetriphenylphosphoranes **35** with isatins and thiourea in refluxing tetrahydrofuran (Figure 6). Some of the synthesized agents revealed promising antimicrobial properties against Gram-positive (*S. aureus* and *B. subtilis*) and Gram-negative (*E. coli* and *P. aeruginosa*) bacteria. The compound with X/Y/Z = F/Cl/H was the most effective hit obtained (zone of growth inhibition = 13.5, 14.0, 8.5, 9.0; 20.3, 26.0, 19.6, 15.6 mm for the effective hit synthesized and Gentamicin against *S. aureus*, *B. subtilis*, *E. coli*, and *P. aeruginos*, respectively) [19].

A variety of spiro[indoline-3,3′-pyrrolidines] **40**–**42** was obtained in excellent yields through azomethine cycloaddition of 3-methyleneoxindolines **32**/**36**/**37** with azomethine ylide (obtained through condensation of paraformaldehyde **38** and sarcosine **39**) in refluxing toluene (Figure 7). Mild to weak antibacterial (Gram-positive, *S. aureus* and Gram-negative, *E. coli*“ MIC = 250–1000 μg/mL) properties were observed by some of the synthesized agents [20].

A series of spirooxindolopyrrolidines **47** and **48** was prepared through dipolar cycloaddition of *β*-nitrostyrenes **43** and azomethine ylides (obtained from the condensation of isatin **44** and tryptophan **45** or *L*-histidine **46**) in different organic solvents (Figure 8). Enhanced/higher yields were observed upon considering ionic liquid ([bmim]Br, 1-butyl-3-methylimidazolium bromide) compared with the conventional solvents. Some of the synthesized agents revealed antifungal properties, of which **47** is the most notable, against *C. albicans* (MIC = 4–16 μg/mL) with inhibition of fungal hyphae and biofilm formation [21].

A series of spiro[indoline-3,4′-[1,3]dithiines] **50** was obtained through a reaction of 3-methyleneoxindolines **36** (obtained from the reaction of the appropriate isatin and malononitrile in MeCN) and 5-(dimercaptomethylene)barbituric or thiobarbituric acid **49** (obtained through the reaction of barbituric/thiobarbituric acid, CS_2_, and triethylamine in MeCN) (Figure 9). The same products were also obtained by utilizing magnetic nanoparticles (Fe3O4@gly@CE) as a recyclable catalyst. Some of the synthesized agents showed considerable inhibitory properties against Gram-positive (*S. fradiae*, *S. pyogenes*, *S. agalactiae*, *S. equinus*, *S. aureus*, *S. epidermidis*, *B. cereus*, *B. thuringiensis*, *R. equi*) and Gram-negative (*A. baumannii*, *P. aeruginosa*, *K. pneumoniae*, *E. coli*, *S. dysenteriae*, *P. mirabilis*, *S. enterica*, *Y. enterocolitica*) bacterial strains, as well as fungal strains (*A. fumigatus*, *C. albicans*, *F. oxysporum*). The synthesized agents with barbituric acid revealed better antimicrobial properties than those with thiobarbituric acid. The most promising agent is that with R = Cl, X = O relative to Gentamicin and Terbinafine (standard antibacterial and antifungal references, respectively; MIC = 8 μg/mL for the synthesized agent and Gentamicin against *E. coli*; MIC = 8, 32 μg/mL for the synthesized agent and Terbinafine against both *A. fumigatus* and *C. albicans*, respectively) [22].

### 3.2. Antimycobacterial Spirooxindoles

Tuberculosis is one of the most severe infectious diseases threatening human life. *Mycobacterium tuberculosis* is a pathogenic bacterial microorganism responsible for infectious diseases. Although different therapeutics have been developed and clinically approved, novel agents are still in demand. This is due to the side effects of the used medications and drug resistance strains discovered [23].

A series of spiro[indoline-3,2′-thiazolidine]-2,4′-diones **52** was prepared through the oxidation reaction of spiroindoles **51** by *m*CPBA in CHCl_3_ (Figure 10). Some of the synthesized agents revealed promising inhibitory properties against MptpB (*M. tuberculosis* protein tyrosine phosphatase B) which possess a controlling role in the immune system useful for the treatment of the disease. The nitro-substituted indole-containing compounds were the most potent hits synthesized (IC_50_ = 1.2 μM of Ar = 3,4-F_2_C_6_H_3_, R = H, R’ = 4-NO_2_) [24].

A variety of spiro[indoline-3,2′-thiazolidines] **54** was synthesized through the microwave-assisted synthetic reaction of isatin-3-imines **53** with thioglycolic acid in DMF using ZrSiO_2_ (Figure 11). Anti-mycobacterial properties against *M. tuberculosis* were noticed for the synthesized agents. The most potent is that of R = NO_2_, X = O (IC_50_ = 12.5 μg/mL) relative to that of Isoniazid (standard reference, IC_50_ = 0.2 μg/mL) [25].

Isoniazid-spirooxindoles **56** were obtained through a reaction of spiro[benzo[*e*]pyrazolo [1,5-*c*][1,3]oxazine-5,3’- indolins] **55** with chloroacetyl chloride and isoniazid (Figure 12). Compound **56** with R= 4-Cl, R’ = Cl is the most effective agent (MIC = 12.5 μg/mL) synthesized relative to isoniazid (MIC = 0.2 μg/mL) against *M. tuberculosis* H 37 Rv [26].

Anti-mycobacterial properties by spiro[indoline-3,2′-[1,3,4]oxadiazols] **30** (Figure 3) against *M. tuberculosis* were also noticed. The most promising are those with halogen-substituted indolyl heterocycle (R = Br and Cl, IC_50_ = 6.25 μg/mL) relative to Ciprofloxacin (IC_50_ = 3.12 μg/mL) [17].

### 3.3. Antiviral Spirooxindoles

The dispiro[indoline-3,2′-pyrrolidine-3′,3″-piperidines] **58** incorporated alkylsulfonyl group attached at the piperidinyl nitrogen were synthesized through regioselective azomethine cycloaddition (obtained through condensation of isatins and sarcosine) with 3,5-bis(ylidene)-4-piperidones **57** (Figure 13). Some of the synthesized agents revealed promising antiviral properties against SARS-CoV-2 with considerable safety behavior towards the host cell (IC_50_ = 7.687, CC_50_ = 262.5 μM, selectivity index = 34.1 for the compound with R = 4-FC_6_H_4_, R’ = Et, R″ = Cl). SARS-CoV-2 is the virus which caused a global pandemic (at the end of 2019), affecting and threatening millions of human lives. Intensive research studies for effective drugs are still one of the hot topics of scientific society [27].

A series of spiro[[1,2,3]triazolo [4,5-*b*]pyridine-7,3′-indolines] **60** was synthesized through reaction of the substituted aminotriazoles **59**, isatin derivatives, and Meldrum’s acid in acetic acid at 100 °C followed by selective *N*-alkylation of the oxindole fragment (Figure 14). Antiviral properties of the synthesized agents were noticed against Dengue virus (DENV) infection (IC_50_ = 0.78, 0.16, 0.035 μM against DENV-1, -2 and -3, respectively, for the compound with R = CH_2_CH_2_CH(Me)_2_). Dengue is a viral disease widely spread in many tropical and subtropical regions. Female *Aedes* mosquitoes are responsible for the distribution of this disease. Fever and pain are the main symptoms, similar to that of flu infection [28].

### 3.4. Anticancer Spiroindoles

Cancer is one of the most deadly diseases threatening several millions of human lives every year. Chemotherapeutical approaches represent one of the major options besides radiotherapy, immunotherapy, and surgery for cancer treatment. Although advances have been achieved in discovering many chemotherapeutical agents, ideal therapeutics (high efficacy with limited side effects) are unreachable. Progress in research directed toward novel bioactive agents is still encouraged [29,30].

The MDM2 (human murine double minute-2) is an important target for cancer therapy. It is a cellular inhibitor for p53 (tumor suppressor). Overexpression of MDM2 was exhibited in many cancer types with wild p53. Due to protein–protein interaction, MDM2 is capable of p53 inhibition (negative regulation through direct binding or ubiquitination/degradation); it is considered a highly attractive target for developing antitumor active agents. The p53 has a circular role in cancer cell apoptosis. In other words, p53 inactivation is an important factor for cancer progression that may be achieved by blocking the interaction of MDM2-p53 [31,32]. Some spiroindole-containing compounds were discovered as MDM2-p53 inhibitors which are entered into human clinical trials (Figure 8) [33].

Azomethine cycloaddition derived from amino acid derivative **61** in MeOH under microwave irradiation at 100 °C with isatin afforded the spirooxindole derivative **62** as a racemic mixture followed by a reductive amination reaction with cyclopropane carboxaldehyde and then *N*-arylation of lactam in Buchwald coupling conditions, with *p*-bromobenzoate giving the final target spirooxindole **63** (Figure 15). Promising efficacy was noticed towards MDM2-p53 and SJSA-1 (p53 wild-type osteosarcoma) (IC_50_ = 161 nM) [34].

A series of spiropyrazoline-oxindole **66** was prepared through nitrilimine cycloaddition of 3-ylidene-2-indolinones **64** with hydrazonyl chlorides **65** in acetonitrile in presence of Et_3_N (sealed tube at 90 °C). Compounds **67** were obtained through the hydrolysis of the corresponding **66** followed by amination (Figure 16). Some of the synthesized analogs revealed promising antiproliferation properties against SJSA-1 (osteosarcoma), LNCaP (prostate), and MCF-7 (breast) cancer cell lines with the ability to dually inhibit MDM2-P53 and MDM4-p53 protein–protein interactions (IC_50_ = 26.1, 219.0; 35.9, 57.4 nM for MDM2-p53, MDM4-p53 corresponding to compounds **67** with R = 6-Cl, R’ = 3-OH, R″ = 4-ClC_6_H_4_, R′′′ = **3**-(3-phenyl-1*H*-pyrazol-1-yl)phenyl and R = 6-Cl, R′ = 3-OH, R″ = 4-ClC_6_H_4_, R′′′ = 3-(1*H*-pyrazol-1-yl)phenyl, respectively) [35].

Spirooxindoles **69** were synthesized through azomethine (obtained from the condensation of isatins with different amino acids in refluxing methanol) cycloaddition with 3-(2-pyrrolidinyl)-2-propen-1-ones **68** (Figure 17). The proposed approach of amino acid to the olefinic linkage was mentioned for the regio- and stereoselectivity of the obtained products. Quantum chemical calculations by DFT (density functional theory) were conducted as supporting elements for the selectivity observations. Antiproliferation properties against diverse human cell lines were explored through the US-NCI program. Some of the synthesized agents revealed MDM2 inhibitory properties [*K*_D_ “MDM_2_ binding by microscale thermophoresis” = 1.13 μM for the most effective agent synthesized with R’ = Cl, R= 3-NO_2_, amino acid = (2*S*,3a*S*,7a*S*)-octahydro-1*H*-indole-2-carboxylic acid] [36].

Analogously, compounds **70** were obtained in a similar synthetic protocol utilizing furyl-containing chalcones. Some of the synthesized agents exhibited potent antitumor properties against MCF-7 (breast) and HepG2 (liver) cancer cell lines (IC_50_ = 4.3, 6.9; 4.7, 11.8, 17.8, 10.3 μM for **70a**, **70b**, and Staurosporine, respectively) [37] (Figure 18).

Nitroisoxazole-containing spirooxindoles **71** were synthesized through the reaction of 5-styrylisoxazole and isatinimines in MeCN in the presence of DBU (Figure 19). Some of the synthesized agents (**71a** and **71b**) exhibited noticeable antitumor properties against MCF-7 cell line with suppression of MDM2-mediated degradation of p53 [38].

Dispirooxindoles **73** were obtained through a cycloaddition reaction of 5-indolidene-2-chalcogen-imidazolones **72** with azmethine ylide (obtained from the condensation of sarcosine and paraformaldehyde) in refluxing toluene (Figure 20). Compound **73** with R = Cl, R’ = 4-MeOC_6_H_4_, X = S revealed considerable antiproliferative properties against HCT116 p53^+/+^ and HCT116 p53^−/−^ CC_50_ = 1.95, 2.35 μM, respectively [39].

Spiro[indoline-3,2’-naphthalenes] **74** were obtained by Michael-aldol cascade reaction of 3-ylideneoxindoles and 2-methyl-3,5-dinitrobenzaldehyde in CH_2_Cl_2_ in the presence of a bifunctional hydrogen-bonding catalyst followed by treatment with HCl/EtOAc (Figure 21). Inhibition of MDM2 and CDK4 in glioblastoma cells were noticed by the synthesized agents. Compound **74** with R = 5-Br, R’ = CO_2_Et is the most potent agent synthesized with IC_50_ = 4.9, 8.6, 9.5 μM against U87MG, U251, and T98G, respectively [40].

Dual inhibitory properties of spirooxindoles bearing sulfonyl function **58** were revealed against VEGFR2 and EGFR [27]. Multi-targeted inhibitors have recently attracted great attention for cancer chemotherapy. This is not only for optimizing effective agents of wide applicability against different types of cancer but also since cancer progression usually depends on several pathways [41]. Some of the synthesized agents revealed high antiproliferative properties (MTT assay). The most potent is that with R = 4-BrC_6_H_4_, R’ = Me, R″ = H (IC_50_ = 3.597, 3.236, 2.434, 12.5 μM against MCF-7 “breast”, HCT116 “colon”, A431 “skin”, and PaCa-2 “pancreatic” cell lines, respectively) relative to 5-fluorouracil (IC_50_ = 3.15, 20.43, 23.44 μM against MCF-7, HCT116, and A431, respectively) and Sunitinib (IC_50_ = 3.97, 9.67, 16.91 μM against MCF-7, HCT116, and PaCa-2 cell lines, respectively) [41].

A series of spirooxindoles **77** was prepared through the reaction of isatins with aroylacetonitriles **75** and 5-aminopyrazole **76** in refluxing AcOH/H_2_O (1:1) (Figure 22). Promising antiproliferative properties were noticed by some of the synthesized agents (IC_50_ = 6.9, 11.8; 0.12, 0.62 μM against HepG2 “liver” and PC3 “prostate” cancer cell lines for the promising agent synthesized “R = H; R’ = Ph” and Doxorubicin, respectively). It has been noticed that the promising agent synthesized exhibited a high pro-apoptotic protein Bax level with low anti-apoptotic protein Bcl-2 in HepG2 cells, confirming its impact on apoptosis induction. The same phenomenon was also supported by testing the caspase-3/9 and p53 protein levels [42]. Additionally, the most promising agents discovered against MDA-MB-231 (triple-negative breast) cancer cell line are those with R/R’ = H/Ph and Cl/Ph (IC_50_ = 6.70 μM for both) relative to Doxorubicin (IC_50_ = 0.12 μM) which also showed good affinity against caspase-3/9 and p53 protein supporting their capability for apoptosis induction [43].

Spiro[chroman-2,3′-indoline] **78** was obtained through the reaction of isatin with 2′-hydroxyacetophenone in EtOH containing Et_2_NH. Meanwhile, the 3-hydroxyindolin-2-one derivative **79** was obtained upon considering 4′-aminoacetophenone instead, which afforded the corresponding imine **80** via condensation with salicylaldehyde. Spiro[indoline-3,3′-pyrazols] **81** were obtained through a reaction of **80** with hydrazines in refluxing EtOH/AcOH (Figure 23). Promising antiproliferative properties were observed by compounds **78** and **81b** (IC_50_ = 0.68, 0.95, 0.74; 1.28, 1.30, 0.76; 2.87, 2.95, 4.76 μM for compounds **78**, **81b**, and Imatinib “standard reference” against MCF-7 “breast”, HepG2 “liver”, and HCT116 “colon” cancer cell lines, respectively) with safe behavior against non-cancer cell line WI-38 (IC_50_ = 204.36, 202.08 μM for compounds **78** and **81b**, respectively). Caspase-3 activation of the promising agents synthesized **78** and **81b** supported the antiproliferation properties revealed. Compound **81b** revealed promising EGFR (epidermal growth factor receptor) inhibitory properties [44].

Ionic liquid [bmim]Br, mediated (100 °C) azomethine (formed from the condensation of tyrosine and isatins) cycloaddition with *β*-nitrostyrenes **43** afforded the spiroindole-pyrrolidines **82** (Figure 24). Some of the synthesized agents revealed mild antiproliferative properties against A549 (alveolar basal epithelial) and Jurkat (acute *T*-cell lymphoma) cancer cell lines (MTT assay) relative to Camptothecin (standard reference) with caspase-dependent (caspase-3) apoptosis (IC_50_ = 38.66, 52.79; 50.88, 51.5 μM against A549, Jurkat for the most promising agent synthesized with R = 4-OMe, R’ = OCF_3_ and Camptothecin, respectively). The chiral configuration of compound **82** was not identified by the authors [45].

Spiro[indole-3,5′-isoxazoles] **84** were obtained through the reaction of 2-arylindoles and nitroalkenes **83** in HCO_2_H containing H_3_PO_4_ at room temperature (Figure 25). Some of the synthesized spiro-containing compounds revealed mild antiproliferative properties against BE(2)-C (neuroblastoma) cell line in the MTT assay (% cell viability = 18 for the most promising agents synthesized with R = 2-thienyl, R′ = H, R″ = Ph; R = 2,3-dihydrobenzo[*b*][1,4]dioxine-6-yl, R′ = H, R″ = 4-MeOC_6_H_4_ at 25 μM) [46].

However, no process of reaction was detected with utilization of nitroalkenes possessing substituent **85** under the same mentioned reaction conditions. As a result, MeSO_3_H was used instead affording a mixture of the corresponding spiroindoles **86** and 3,3′-bis(1*H*-indole)methane derivatives **87** [46] (Figure 26).

Azomethine cycloaddition of equimolar amounts of chalcones **88**, 2-(piperazin-1-yl)ethanamine and isatin in stirring ethanol afforded the corresponding spiro[indoline -3,2′-pyrrolidins] **89** (Figure 27). The chiral configuration of pyrrolidine ring was not identified and reported by the authors. Some of the synthesized agents revealed considerable antiproliferative properties (MTT assay) against KB cell line (the most potent is the compound with R = Me, IC_50_ = 6.5 μM) [47].

Thiazolo-pyrrolidine-spirooxoindoles **91** were analogously obtained in good to excellent yields (71–89%) through multicomponent azomethine (obtained from the condensation of isatin and *L*-thioproline) reaction with indolyl-containing chalcones **90** in boiling MeOH (Figure 28). The most effective antiproliferative agent is that with R = 4-CF_3_C_6_H_4_ (IC_50_ = 7.0, 5.5 μM against HCT116 “colon” and HepG2 “liver” cancer cell lines, respectively) relative to that of Cisplatin (IC_50_ = 12.6, 5.5 μM against HCT116 and HepG2 cancer cell lines, respectively) [48].

The spirooxindole-pyrrolo-carbazoles **93** were obtained through azomethine (obtained through condensation of benzylamine and isatin) cycloaddition with 2-ylidene-1*H*-carbazole-1-ones **92** in refluxing dioxane/methanol (1:1) (Figure 29). Antiproliferative properties were noticed for the synthesized agents. The most potent are those derived from 2-thienylidene-1 *H*-carbazole-1-ones (IC_50_ = 14, 13, 15; 15, 14, 16 μM for compounds with R’ = Me, Cl, and H relative to Cisplatin; IC_50_ = 9, 10 μM against MCF-7 “breast” and A-549 “lung” cancer cell lines, respectively) [49].

A series of spiro[indoline-pyrrolizin]-ones **95** was obtained through azomethine (obtained through isatins and *L*-proline) cycloaddition with 3-(benzo[7]annulen-8-yl)-2-propen-1-ones **94** in refluxing methanol (Figure 30). Some of the prepared agents exhibited promising antiproliferative properties against SKNSH (neuroblastoma) cancer cell lines. The most promising agents are those with R/R’/R″ = OMe/H/Cl; OMe/H/I (IC_50_ = 4.61 and 5.04 μM, respectively) relative to Doxorubicin (IC_50_ = 6.3 μM) [50].

Azomethine ylides (generated from the condensation of isatins and 2-*S*-octahydro-1*H*-indole-2-carboxylic acid) with 2,6-bis(arylidene)cyclohexanones **96** afforded the corresponding spirooxindoles **97** in good yields (Figure 31). Some of the prepared agents exhibited mild antiproliferative properties against PC3 (prostate), Hela (cervical), and (MCF-7, MDA-MB231) breast cancer cell lines in MTT assay relative to Doxorubicin (standard reference) (IC_50_ = 3.7 μM for the compound with R = H, R′ = 6-Cl against PC_3_, IC_50_ = 7.1 μM for the compound with R = R′ = H against HeLa relative to IC_50_ = 1.9, 0.9 μM for Doxorubicin against PC_3_ and HeLa, respectively) [51].

Ultrasonic irradiation of isatins, phenacyl bromides, and phenacylidenetriphenylphosphorane **98** in water containing Et_3_N at 20–30 °C (60 W) afforded the corresponding spirocyclopropaneoxindoles **99** in high yields (Figure 32). Some of the synthesized compounds revealed promising antiproliferation properties against HeLa (cervical) cancer cell line in the MTT assay (IC_50_ = 9.30, 4.50; 6.33, 1.86 for the most effective agent synthesized with R = H, R′ = Cl, R″ = [1-(4-bromobenzyl)-1*H*-1,2,3-triazol-4-yl] and Doxorubicin at 24 h and 48 h, respectively) [52].

A series of spiro[chromeno[4,3-*b*]chromene-7,3′-indolines] **100** and spiro[indeno[2′,1′:5,6]pyrano[3,2-*c*]chromene-7,3′-indolines] **101** were obtained through eco-friendly synthetic approach through the reaction of isatins, 4-hydroxycoumarin, and 5,5-dimethylcyclohexande-1,3-dione or 1*H*-indene-1,3(2*H*)-dione, respectively, in H_2_O in the presence of *p*-toluenesulfonic acid (*p*-TSA.H_2_O) at room temperature (Figure 33). Some of the synthesized agents revealed promising antiproliferation properties against (PC-3 and LNCaP) prostate cancer cell lines and alkaline phosphatase inhibitory properties. The most potent agents discovered are **101** derived from 5-bromoisatine (R = H, R′ = Br; IC_50_ = 0.025 μM) and *N*-allylisatin (R = allyl, R′ = H; IC_50_ = 0.25 μM) relative to Bicalutamide (standard reference, IC_50_ = 1.25, 1.50 μM) against PC3 and LNCaP cell lines, respectively [53].

A series of spiro[acridine-9,3′-indolines]-1,2′,8-triones **102** was obtained through multicomponent free solvent reaction (grinding for 3–4 min) of isatin, 1,3-cyclohexanedione, and the appropriate aromatic amine in presence of *p*-toluenesulfonic acid as a catalyst (Figure 34). Some of the synthesized compounds revealed promising antiproliferative properties against MCF-7 (breast) cancer cell line (MTT assay). The most promising agent was that of R = 4-MeOC_6_H_4_ relative to that of Doxorubicin (GI_50_/TGI/LC_50_ = 0.01/0.02/0.71, 0.02/0.21/0.74 μM for the promising synthesized agent and Doxorubicin, respectively) [54].

Spiro[chromene-2,3′-indolin] **104** was obtained through the reaction of 5-(morpholinosulfonyl)isatin **103** with 2′-hydroxyacetophenone in a two-step reaction through addition of methanol containing Et_2_NH followed by heating (95 °C) in AcOH containing a catalytic amount of HCl. Meanwhile, spirooxindoles **105** and **106** were obtained through reaction of **103** with malononitrile and either pyrazol-5-one or phenols in refluxing methanol containing a catalytic amount of piperidine or NaOAc. In an alternative pathway, compounds **105** and **106** were obtained through reaction of ylidenemalononitrile **107** with either pyrazol-5-one or phenols under the same reaction conditions (Figure 35). Mild antiproliferative properties were revealed by the synthesized spiro-compounds against HepG-2 (liver), HCT-116 (colon) and MCF-7 (breast) cancer cell lines relative to Doxorubicin (MTT assay) (IC_50_ = 29.05, 25.31, 33.75; 5.59, 7.03, 4.89 μM against HepG-2, HCT-116, and MCF-7 for compound **97** with R = Ph and Doxorubicin, respectively) [55].

Chromium oxide-promoted oxidation of *N*-[(1-methoxyindol-2-yl)methyl]-N’-(aryl)thioureas **108** afforded the corresponding spiro[indoline-2,5′-[4′,5′]dihydrothiazoles] **109** (Figure 36). Some of the synthesized agents revealed mild antiproliferative properties against HCT116 (colon), Jurkat (leukemic T cell lymphoma), and MCF-7 (breast) cancer cell lines relative to Cisplatin (standard reference) (IC_50_ = 33.7, 35.5, 36.9; 15.3, 16.2, 15.6 μM for the synthesized agent with R = 4-CF_3_ and Cisplatin against HCT116, Jurkat, and MCF-7, respectively) [56].

A group of spirooxindoles **113** was obtained through the reaction of phenacyl bromides **112** with [5-mercapto-1,2,4-triazole-4-ylimino]-2-indolinones **111** in refluxing methanol containing Et_3_N. The latter were prepared via condensation of isatin with the appropriate 4-amino-1,2,4-triazol-5-thiols **110** in refluxing MeOH containing a catalytic amount of *p*-toluenesulfonic acid (Figure 37). Some of the targeted spiro-analogs revealed promising antiproliferative properties against MGC803, a human gastric cell line (IC_50_ = 9.49 μM) relative to that of 5-fluorouracil (IC_50_ = 25.54 μM) in MTT assay [57].

Ylideneoxindoline-2-ones **32** in ethanol containing Et_3_N at room temperature afforded the corresponding spiro[indoline-3,3′-pyrrolidines] **114** (Figure 38). Some of the synthesized agents revealed noticeable antiproliferative properties (MTT assay) against HepG2 (liver) and CT26 (colon) cancer cell lines (% cell death = 15.3, 35.39; 50.89, 75.17 for the compound with R = CH_2_Ph, R′ = F, R″ = 4-MeC_6_H_4_ against HepG2 and CT26 at 50 μg/mL, respectively) [58].

### 3.5. Antimalarial Spirooxindoles

Malaria is one of the most endemic diseases worldwide. This is due to the suitable environment for mosquitoes in tropical and subtropical regions with a high global population. Many parasitic species of protozoa causing this disease have been identified as transmitted to humans through mosquito bites. Although several agents were investigated against malaria (Artemisinin **115**, Nobel Prize in Physiology awarded to Professor Youyou Tu due to efforts in its discovery) [59] (Figure 9), there remains a need for newer ones. This is attributed to the resistance observed by some varieties of this parasite [60].

Artemisinin and its derivatives are fast-acting agents against the asexual blood stage parasites. Co-administration of artemisinin analog (fast-acting) with a long-acting drug as first-line therapeutics is recommended [61]. Cipargamin (Figure 2) is a promising antimalarial compound in clinical studies as a therapeutic inhibiting blood-stage *P. falciparum*. This is considered a promising agent to combat the artemisinin resistance parasite [62]. Hepatic safety behavior was achieved through clinical studies (phase II) across wide-range doses [63,64].

A series of spiro[indoline-3,2′-[1,3,4]oxadiazols] **117** were obtained through nitrilimine cycloaddition obtained by dehydrochlorination of hydrazonyl chloride **116** with isatins in CH_2_Cl_2_ containing Et_3_N at room temperature (Figure 39). Some of the synthesized spirooxindoles reveal promising properties against erythrocytic stage of *P. falciparum* and the liver-stage of *P. berghei*. This supports the possibility of developing active agents inhibiting both blood-stage and (*P. falciparum*) and liver-stage (*P. berghei*) parasites [65].

Spiro[indoline-3,2′-quinolins] **121** and spiro[indoline-3,5′-pyrano[3,2-*c*]quinolins] **122** were obtained through Povarov reaction taking place from imines **118** (formed from the condensation of substituted anilines and isatin) and alkene-containing compound, *trans*-isoeugenol **119** or 3,4-dihydro-2*H*-pyran **120**, respectively in CH_2_Cl_2_ in presence of BF_3_.OEt_2_ (Lewis acid) at room temperature (Figure 40). Some of the synthesized **121** revealed efficacy against *P. falciparum* drug-resistant FCR-3 strain (IC_50_ = 1.52–4.20 μM) relative to that of chloroquine (IC_50_ = 0.11 μM) through in vivo testing. Meanwhile, compounds **122** revealed activity (IC_50_ = 1.31–1.80 μM) against *P. falciparum* drug-sensitive 3D7 strain relative to that of chloroquine (IC_50_ = 0.0127 μM) [66].

Mild anti-plasmodial properties were observed by spirooxindoles **124a** and **124b**, revealing weak properties against the artemisinin-sensitive and resistant *P. falciparum* strains. Spirooxindoles **124a/b** were obtained through a reaction of indolo[8,7-*b*]indolizine **123** with *N*-bromosuccinimide (NBS) in AcOH/THF/H_2_O (1:1:1) at 0 °C to room temperature [67] (Figure 41).

### 3.6. Anti-Inflammatory Spirooxindoles

Spirooxindoles **126** were prepared via azomethine (obtained through condensation of isatin and pipecolinic acid) cycloaddition of 3,5-bis(ylidene)-4-piperidones **125** in ionic liquid “[bmim]Br, (1-butyl-3-methylimidazoliumbromide)” at 100 °C (Figure 42). Some of the synthesized agents showed promising acute and chronic anti-inflammatory properties relative to that of indomethacin with inhibitory observations against PGE_2_, TNF-*α*, and nitrite levels (% reduction of TNF-*α* = 36.23, 38.52, 39.19, 37.13 and % decline in nitrite level = 42.99, 41.22, 44.04, 41.64 for the synthesized compounds with R = 4-MeC_6_H_4_, 4-MeOC_6_H_4_, 3-NO_2_C_6_H_4_, and indomethacin, respectively) [68].

### 3.7. Antihyperglycemic Spirooxindoles

Multi-component azomethine (formed from isatin and thioproline) cycloaddition with 4-arylidene-5(4*H*)-oxazolones **127** in refluxing methanol afforded the corresponding spirooxindoles **128** and **129** as a mixture of two diastereoisomers (Figure 43). Promising antihyperglycemic properties were observed by some of the synthesized agents. Compound **128** with R = 4-MeC_6_H_4_ is the most potent (IC_50_ = 1.76, 4.81 μM against *α*-amylase from human saliva and *α*-glucosidase from *Saccharomyces cerevisiae*, respectively) [69].

Spirooxindoles **130** were obtained through azomethine (obtained through condensation of isatin and benzylamine) cycloaddition with chalcones **88** in refluxing methanol (Figure 44). Some of the synthesized agents revealed AGE (advanced glycation end, which is the formation of sugar-derived substances) inhibitory properties in the BSA (bovine serum albumin) glucose assay, supporting the suitability for diabetes. The most promising agent synthesized is that with R = Ph (IC_50_ = 11.37 μM) relative to the aminoguanidine “standard reference” (IC_50_ = 40.54 μM) [70].

### 3.8. Anti-Leishmanial Spirooxindoles

Spiro[indoline-3,2′-quinolins] **121** (Figure 10) obtained through Diels–Alder reaction of imines and *trans*-isoeugenol (Figure 40) revealed promising anti-leishmanial properties. The most promising is that with R/R’ = Et/H against *L. braziliensis*, with safe behavior towards mammalian cells viability [71].

Spirooxindoles **131** were prepared through cycloaddition of 2,6-bis(ylidend)cyclohexanones and azomethine ylide (formed from the condensation of proline and isatin) (Figure 45). Few of the synthesized agents revealed anti-leishmanial properties (a compound with Ar = 3-NO_2_C_6_H_4_ is the most effective agent with IC_50_ = 6.8 μg/mL relative to amphotericin B (IC_50_ = 0.29 μg/mL) [72].

## 4. Conclusions and Future Directions

Development of new potential therapeutics is always a challenge for medicinal chemistry research. Spirooxindoles represent an important class of heterocyclic compounds and have emerged as attractive scaffolds with unique structural architecture and diverse pharmacological properties. Many natural and synthetic compounds have been identified as potential pharmacophores. Even though there have been several important breakthroughs and encouraging results on spirooxindoles as potential therapeutic agents as discussed above, challenges and opportunities remain for medicinal chemistry research. Several investigations on spirooxindole scaffolds were reported and studied in recent years [73,74,75,76,77]. The current compiled synthetic protocols of pharmacologically active spirooxindole scaffolds will provide an efficient platform to create a new generation of potential spirooxindole analogues for various diseases.

## Data Availability

Not applicable.

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
