# Peer review of "Spirooxindole: A Versatile Biologically Active Heterocyclic Scaffold"

_molecules, 2023, doi:10.3390/molecules28020618_

Round 1

Reviewer 1 Report

The review comprises of recent reports of spirooxindoles, major focus of the review was given to pharmacological importance of spirooxindole in natural products, antibacterial, antimicrobial, anticancer, antifungal, antimycobacterial, antileishmanial etc. and other areas. Most potent compounds were listed according to their IC50 & MIC values along with their synthetic routes. The article is covered most of the interesting recent literature, however it needs more improvements with respective to discussion of the work in detail and future directions in the area. Also, the conclusion section should be improved. The review article is suitable for the publication after the major revision. Yet, the author needs to focus on few points as mentioned below:

·       The author should cross verify the title of the review as “spirooxindole is more suitable than spiroindole” which would be better fit for the review article. In the entire article it should be replaced including conclusion.

·       The spirooxindole in Fig 5 chiral bonds were shown in different representations (doted), from one structure to another. The author could represent them in one similar format over entire article.

·       In scheme no-7,11,36,24 the spiro structures were not drawn uniformly. The structures were also seen to be scaled to small size in chemdraw by which atoms are looking bit different. The author could re-draw the structures accordingly.

·       In scheme no. 3,5,6,7,8,9,13,14,15,18,19,25,26,29,30,32,35,36, - The authors should mention IC50 values of the corresponding literature reports of potent molecules. It was necessary, as review is focused on medicinal chemistry and pharmacological importance of spirooxindoles.

·       The reference style is not matching according to journal format and make it according to the journal guidelines.

·       The authors should discuss the following recent articles of spirooxindoles and its pharmacological significance:

https://doi.org/10.1002/ejoc.202001261

DOI: 10.1016/j.ejmech.2015.09.025

https://doi.org/10.1021/acs.joc.2c00863

https://doi.org/10.1021/acsomega.2c03790

Author Response

Date:               December 27, 2023

Subject:           Revised Manuscript ID molecules-2080614 

Dear Reviewer,

Thank you for reviewing our manuscript. We appreciate your helpful and valuable suggestions. We have reviewed them carefully and believe we have made all appropriate changes to our manuscript.

Comments and Suggestions for Authors:

The review comprises of recent reports of spirooxindoles, major focus of the review was given to pharmacological importance of spirooxindole in natural products, antibacterial, antimicrobial, anticancer, antifungal, antimycobacterial, antileishmanial etc. and other areas. Most potent compounds were listed according to their IC50 & MIC values along with their synthetic routes. The article is covered most of the interesting recent literature, however it needs more improvements with respective to discussion of the work in detail and future directions in the area. Also, the conclusion section should be improved. The review article is suitable for the publication after the major revision. Yet, the author needs to focus on few points as mentioned below:

Comment 1

The author should cross verify the title of the review as “spirooxindole is more suitable than spiroindole” which would be better fit for the review article. In the entire article it should be replaced including conclusion.

Response

Thank you for your suggestion and as suggested we have now changed to spirooxindole

Comment 2

The spirooxindole in Fig 5 chiral bonds were shown in different representations (doted), from one structure to another. The author could represent them in one similar format over entire article.

Response

We have now corrected the error.

 Comment 3

In scheme no-7,11,36,24 the spiro structures were not drawn uniformly. The structures were also seen to be scaled to small size in chemdraw by which atoms are looking bit different. The author could re-draw the structures accordingly.

Response

We have now carefully checked and corrected the structures.

Comment 4

In scheme no. 3,5,6,7,8,9,13,14,15,18,19,25,26,29,30,32,35,36, - The authors should mention IC50 values of the corresponding literature reports of potent molecules. It was necessary, as review is focused on medicinal chemistry and pharmacological importance of spirooxindoles.

Response

We have now included the missing pharmacological information.

Comment 5

The reference style is not matching according to journal format and make it according to the journal guidelines.

Response

Thank you, we have now updated the reference format as per the journal style.

Comment 6

The authors should discuss the following recent articles of spirooxindoles and its pharmacological significance:

https://doi.org/10.1002/ejoc.202001261

DOI: 10.1016/j.ejmech.2015.09.025

https://doi.org/10.1021/acs.joc.2c00863

https://doi.org/10.1021/acsomega.2c03790

Response

We have now included the suggested reference in the manuscript.

Reviewer 2 Report

The manuscript proposed by S. S. Panda  and collaborators titled « Spiroindole : A versatile Biologically Active Heterocyclic Scaffold» and submitted to Molecules (molecules-2080614-peer-review-v1), summarizes the recent publications (last five years 2018-2022) describing both natural and synthetic spirooxindole-containing compounds. Starting with spirooxindole -compounds isolated from plants and microorganisms (around 6 references), the authors have then chosen to describe synthetic spirooxindole-compounds, classified in nine sections, based on specific biological properties (Antibacterial (8 references), antimycobacterial (2 references), antiviral (2 references), anticancer (20 references), antimalarial (3 references), analgesic,  anti-inflammatory (1 reference), anti-diabetic (2 references), antileshmania (1 reference). For each series of compounds the chemical synthesis is briefly described and dicussed, as well as the biological results obtained with the most promising compounds.  

It must be noted that  all the natural or synthetic compounds described in this review are 2-oxoindoles. Therefore for each chemical synthesis (except 2 examples), isatin (or a derivative of isatin) can be identified as a main precursor. This is a restriction that the authors could precise in the Title (Spirooxindole) and Abstract ( … and synthetic spirooxindole-containing compounds prepared from isatin or isatin-derivatives, and reported in the last five years.).

Taking these remarks in consideration, the reviewer has noted that recent publications (not exhaustive list !) describing the synthesis of novel  spirooxindole-compounds  with specific antitumor (for example : European Journal of Medicinal Chemistry (2021), 217, 113359 ; Molecules (2021), 26(24), 7645 ; Acta Pharmaceutica Sinica B (2020), 10(8), 1492-1510, Molecules (2020), 25(23), 5581 ; … ), anti-bacterial (for example :  Medicinal Chemistry Research (2022), 31(6), 1026-1034 ; Journal of Molecular Structure (2020), 1222, 128881, …), antiviral (for example : European Journal of Medicinal Chemistry (2021), 217, 113359, ), antiinflammatory/antileishmanial (for example : ChemistrySelect (2019), 4(35), 10510-10516 ) properties,  seem not to be  cited in this review .  

Could the authors justify their selection of the literature works describing synthesis and activity of spirooxindole-containing compounds, in the last five years ? Or complete the manuscript with missing literature data as a major revision?

General comments :

1.       Change spiroindole for spirooxindole, in the manuscript

2.       Sometimes biological results of the « most promising compounds » are given and sometimes no. This must be completed because such informations are important for the reader (see lines : 110, 123, 152, 222, 228, 332, 408, 439)

Specific comments :

11.       Fig 4, compound 5 (isospeciofoline) is compound 6 ; compound 7 is Mitragynine oxindole B

22.       Line 73 : delete « Scheme 15 »

33.       Scheme 1 : Wittig reagent preparation description is not necessary but must be writen as in scheme 5 to keep homogeneity

44.       Scheme 3 : double bond is missing in the indole ring

55.       Scheme 7 could be simplified, deleting the C-H bonds (on 46 and 47) and diastereoselectivity could be precised,as well as biological values for 47, in the text (C Albicans MIC 4-16 mg/mL).

66.       Lines 145-143 : what is the magnetic supported catalyst ? precise barbiturica acid 48.

77.       Line 166 : precision for nitro compound ; R = NO2 or R’=NO2 ?

88.       Line 192 : …selective N-alkylation of the oxindole fragment

99.       Line 206 - 215: precise MDM2 ligase and p53 protein, the ref 29 is about a PROTAC and not PPI-inhibitors of MDM2-p53

110.   Lines 218-221 : is it a diastereomeric mixture ? …followed by a reductive amination reaction with cyclopropane carboxaldehyde and then N-arylation of lactam  in Buchwald coupling conditions.

111.   Line 229 : …to dually inhibit MDM2-P53 and MDM4-p53 protein-protein intractions.

112.   Line 234 : how to explain regio- and stereoselectivity giving 66 ?

113.   Line 254 : 5-fluorouracil

114.   Scheme 17 : Et2NH is missing in the condensation with acetophenone

115.   Scheme 18 : Is it racemic tyrosine ??For compound 74, carbons configuration are not clear

116.   Line 288 : « However, with nitroalkenes possessing an a-substituent 77.. ». Scheme 20 , a nitrogen atom is missing in 79 ?

117.   Scheme 81 : what about the relative configurations of the carbons of the pyrrolidin skeleton in 81 ?.

118.   Lines 335-342 : compounds 92 are spiro indoline triones. Correct isatin (line 335. Most potent compound 92 is for R = NO2 and R = ?, same remark for 93 R’ = Br and R= ?.

119.   Scheme 28 : title : Synthesis of spiro[acridine 9,2’-indolines]-1,3,8-triones

220.   Scheme 30 : is the unique exemple giving 3-oxindole derivatives , not starting from isatin. This reference (50) is perhaps not crucial in this review ?

221.   Paragraphs 3.5 and 3.9 could be fused, because both concern parasite infections ? The synthetic pathway to access to compounds 113, must also be described.

222.   Line 405 : Nitrile imine cycloaddition (obtained by dechlorination of hydrazonyl chloride 108 )

223.   Schem 33 : title : Synthesis of spiro[indoline-3,2’-[1,3,4]-oxadiazol]-2-ones

224.   Lines 413- : this is a Povarov-reaction between an aromatic imine and an alkene, catalysed with L.A ;

225.   Scheme 35 described spiroindole-2-ones not prepared from isatin but by ring-contraction. This exemple could be deleted

226.   Item 3.6 can be deleted, because doesn’t describe oxindole ?

 In conclusion, this review can be accepted for publication in Molecules, but with some corrections and as a major revision, completing the manuscript with the suggested omitted literature  data and publications.

Author Response

Date:               December 27, 2023

Subject:           Revised Manuscript ID molecules-2080614 

Dear Reviewer,

Thank you for reviewing our manuscript. We appreciate your helpful and valuable suggestions. We have reviewed them carefully and believe we have made all appropriate changes to our manuscript.

Comments and Suggestions for Authors:

The manuscript proposed by S. S. Panda  and collaborators titled « Spiroindole : A versatile Biologically Active Heterocyclic Scaffold» and submitted to Molecules (molecules-2080614-peer-review-v1), summarizes the recent publications (last five years 2018-2022) describing both natural and synthetic spirooxindole-containing compounds. Starting with spirooxindole -compounds isolated from plants and microorganisms (around 6 references), the authors have then chosen to describe synthetic spirooxindole-compounds, classified in nine sections, based on specific biological properties (Antibacterial (8 references), antimycobacterial (2 references), antiviral (2 references), anticancer (20 references), antimalarial (3 references), analgesic,  anti-inflammatory (1 reference), anti-diabetic (2 references), antileshmania (1 reference). For each series of compounds the chemical synthesis is briefly described and dicussed, as well as the biological results obtained with the most promising compounds.

Comment 1

It must be noted that all the natural or synthetic compounds described in this review are 2-oxoindoles. Therefore for each chemical synthesis (except 2 examples), isatin (or a derivative of isatin) can be identified as a main precursor. This is a restriction that the authors could precise in the Title (Spirooxindole) and Abstract ( … and synthetic spirooxindole-containing compounds prepared from isatin or isatin-derivatives, and reported in the last five years.).

Response

Thank you for your suggestion and as suggested we have now updated the title.

Comment 2

Taking these remarks in consideration, the reviewer has noted that recent publications (not exhaustive list !) describing the synthesis of novel  spirooxindole-compounds  with specific antitumor (for example : European Journal of Medicinal Chemistry (2021), 217, 113359 ; Molecules (2021), 26(24), 7645 ; Acta Pharmaceutica Sinica B (2020), 10(8), 1492-1510, Molecules (2020), 25(23), 5581 ; … ), anti-bacterial (for example :  Medicinal Chemistry Research (2022), 31(6), 1026-1034 ; Journal of Molecular Structure (2020), 1222, 128881, …), antiviral (for example : European Journal of Medicinal Chemistry (2021), 217, 113359, ), antiinflammatory/antileishmanial (for example : ChemistrySelect (2019), 4(35), 10510-10516 ) properties,  seem not to be  cited in this review.

Response

We have now included the suggested references in the manuscript.

Comment 3

Could the authors justify their selection of the literature works describing synthesis and activity of spirooxindole-containing compounds, in the last five years ? Or complete the manuscript with missing literature data as a major revision?

Response

We published a review article on the same topic in 2017 (ref. 7) so now we are extended our search and prepared the current manuscript based on representatives having promising/potent bio-properties.

Comment 4

Change spiroindole for spirooxindole, in the manuscript.

Response

We have now changed.

Comment 5

  Sometimes biological results of the « most promising compounds » are given and sometimes no. This must be completed because such informations are important for the reader (see lines : 110, 123, 152, 222, 228, 332, 408, 439)

Response

We have now included the missing pharmacological information.

Comment 6

Fig 4, compound 5 (isospeciofoline) is compound 6 ; compound 7 is Mitragynine oxindole B

Response

We have now corrected the error. Thank you

Comment 7

Line 73 : delete « Scheme 15 »   

Response

We have now corrected the error.

Comment 8

Scheme 1 : Wittig reagent preparation description is not necessary but must be writen as in scheme 5 to keep homogeneity.

Response

We have now updated it as per the suggestion.

Comment 9

Scheme 3 : double bond is missing in the indole ring.

Response

We have now corrected the error.

Comment 10

Scheme 7 could be simplified, deleting the C-H bonds (on 46 and 47) and diastereoselectivity could be precised,as well as biological values for 47, in the text (C Albicans MIC 4-16 mg/mL).

Response

We have now updated the scheme as suggested and included the missing information.

Comment 11

Lines 145-143 : what is the magnetic supported catalyst ? precise barbiturica acid 48.

Response

It has been added to the text (Fe3O4@gly@CE) of the article.

Comment 12

Line 166 : precision for nitro compound ; R = NO2 or R’=NO2 ?

Response

The reference “Tetrahedron 67, 6713‒6729 (2011)” has been checked and the structure is correct..

Comment 13

Line 192 : …selective N-alkylation of the oxindole fragment.

Response

The suggested information is now updated.

Comment 14

Line 206 - 215: precise MDM2 ligase and p53 protein, the ref 29 is about a PROTAC and not PPI-inhibitors of MDM2-p53

Response

We have now updated the information with a new reference “Eur. J. Med. Chem. 236, 114334 (2022). https://doi.org/10.1016/j.ejmech.2022.114334”.

Comment 15

Lines 218-221 : is it a diastereomeric mixture ? …followed by a reductive amination reaction with cyclopropane carboxaldehyde and then N-arylation of lactam  in Buchwald coupling conditions.

Response

We have now revised the statement for better understanding.

Comment 16

Line 229 : …to dually inhibit MDM2-P53 and MDM4-p53 protein-protein intractions.

Response:

We have now updated the sentence.

Comment 17

Line 234 : how to explain regio- and stereoselectivity giving 66 ?

Response

A statement is now included “Proposed approach of amino acid to the olefinic linkage was mentioned for the regio- and stereoselectivity for the obtained products. Quantum chemical calculations by DFT (density functional theory) were conducted as supporting elements for the selectivity observations” 

Comment 18

Line 254 : 5-fluorouracil

Response

We have now corrected the error.

Comment 19

Scheme 17 : Et2NH is missing in the condensation with acetophenone

Response

We have now updated with the missing information.

Comment 20

Scheme 18 : Is it racemic tyrosine ??For compound 74, carbons configuration are not clear

Response

The original reference (Bioorg. Med. Chem. 27 (2019) 2487–2498” did not specify the configuration. We have now included the information in the manuscript.

Comment 21

Line 288 : « However, with nitroalkenes possessing an a-substituent 77.. ». Scheme 20 , a nitrogen atom is missing in 79 ?

Response

We have now corrected the error.

Comment 21

Scheme 81 : what about the relative configurations of the carbons of the pyrrolidin skeleton in 81 ?.

Response

The information is missing in the original reference “Journal of Molecular Structure 1267 (2022) 133551”. We have now included the information in the manuscript.

Comment 22

Lines 335-342 : compounds 92 are spiro indoline triones. Correct isatin (line 335. Most potent compound 92 is for R = NO2 and R = ?, same remark for 93 R’ = Br and R= ?.

Response

We do apologize and have now corrected the error.

Comment 23

Scheme 28 : title : Synthesis of spiro[acridine 9,2’-indolines]-1,3,8-triones

Response

We have now updated.

Comment 24

Scheme 30 : is the unique exemple giving 3-oxindole derivatives , not starting from isatin. This reference (50) is perhaps not crucial in this review ?

Response

We do agree with your point but due to the oxidation process, 2-spiroindolin-3-one was formed as the product.

Comment 25

Paragraphs 3.5 and 3.9 could be fused, because both concern parasite infections ? The synthetic pathway to access to compounds 113, must also be described.

Response

We do agree that both 3.5 and 3.9 are parasite infections but they both belongs to two different major diseases. We you still suggest we will combine them

We have now described the synthetic pathway in the text “Pavarov reaction taking place from to imines 117 (formed from the condensation of substituted anilines and isatin) and alkene-containing compound, trans-isoeugenol 118 or 3,4-dihydro-2H-pyran 119, respectively in CH2Cl2 in presence of BF3.OEt2 (lewis acid) at room temperature”.

Comment 26

Line 405 : Nitrile imine cycloaddition (obtained by dechlorination of hydrazonyl chloride 108 )

Response

We have now updated with the suggested information.

Comment 27

Schem 33 : title : Synthesis of spiro[indoline-3,2’-[1,3,4]-oxadiazol]-2-ones

Response

We have now revised as suggested.

Comment 28

Lines 413- : this is a Povarov-reaction between an aromatic imine and an alkene, catalysed with L.A ;

Response

This information is now included.

Comment 29

Scheme 35 described spiroindole-2-ones not prepared from isatin but by ring-contraction. This exemple could be deleted

Response

As the end product is the spirooxindoles, we believe this is relevant. If you still suggest it, we will remove from the manuscript.

Comment 30

Item 3.6 can be deleted, because doesn’t describe oxindole ?

Response

We have now deleted the compound as suggested.

Reviewer 3 Report

The review article entitled “Spiroindole: A Versatile Biologically Active Heterocyclic Scaffold” by Panda et al reported the recent development of both natural and synthetic spiroindole-containing compounds in the last five years. This review is useful for synthetic and medicinal chemists as it provides the compiled synthetic protocols of pharmacologically active spriroindole scaffolds for the development of drug candidates for various diseases. Although this review is important, it needs to be revised thoroughly as there are lots of grammar mistakes and lots of image format problem needs to be changed. Some of the following points are appended below.

(1) Page 2, line 37, Mistake in the label of Fig.2. the effective agent is SOID-8 instead of SOID-6.

(2) Page 3, line 54, Numbering mistake in Fig.4.

(3) Page 4, line 84, change the sentence from “The latter due to interaction with 20 furnished finally the spiroindoles 21” to “The latter due to the interaction with 20 finally furnished the spiroindoles 21”.

(4) Page 5, Scheme 1, It is better to unify whether the substrate should be numbered or not.

(5) Page 5, line 98, change “a” to “the”.

(6) Page 5, line 106, change “were” to “was”.

(7) Page 7, line 135, change “solvent” to “solvents”.

(8) Page 9, line 166, change “are” to “were”.

(9) Page 10, line 191, change “derivative” to “derivatives”.

(10) It is better to unify the font size of label in all of the scheme.

Author Response

Date:               December 27, 2023

Subject:           Revised Manuscript ID molecules-2080614 

Dear Reviewer,

Thank you for reviewing our manuscript. We appreciate your helpful and valuable suggestions. We have reviewed them carefully and believe we have made all appropriate changes to our manuscript.

Comments and Suggestions for Authors:

The review article entitled “Spiroindole: A Versatile Biologically Active Heterocyclic Scaffold” by Panda et al reported the recent development of both natural and synthetic spiroindole-containing compounds in the last five years. This review is useful for synthetic and medicinal chemists as it provides the compiled synthetic protocols of pharmacologically active spriroindole scaffolds for the development of drug candidates for various diseases. Although this review is important, it needs to be revised thoroughly as there are lots of grammar mistakes and lots of image format problem needs to be changed. Some of the following points are appended below. 

Response

We have now carefully checked an corrected the errors.

Comment 1

Page 2, line 37, Mistake in the label of Fig.2. the effective agent is SOID-8 instead of SOID-6.

Response

We have now corrected the error and updated the figure.

Comment 2

Page 3, line 54, Numbering mistake in Fig.4.

Response

Thank you, we have now corrected the error.

Comment 3

Page 4, line 84, change the sentence from “The latter due to interaction with 20 furnished finally the spiroindoles 21” to “The latter due to the interaction with 20 finally furnished the spiroindoles 21”.

Response

We have now updated the information.

Comment 4

Page 5, Scheme 1, It is better to unify whether the substrate should be numbered or not.

Response

We have now revised all schemes for uniformity.

Comment 5

Page 5, line 98, change “a” to “the”.

Response

It has been corrected.

Comment 6

Page 5, line 106, change “were” to “was”.

Response

It has been corrected.

Comment 7

Page 7, line 135, change “solvent” to “solvents”.

Response

It has been corrected.

Comment 8

Page 9, line 166, change “are” to “were”.

Response

It has been corrected.

Comment 9

Page 10, line 191, change “derivative” to “derivatives”.

Response

The reaction utilized only one isatin derivative.

Comment 10

It is better to unify the font size of label in all of the scheme.

Response

We have now updated the schemes as suggested.

Round 2

Reviewer 3 Report

Accept in present form